# MRI-Based Radiomics Input for Prediction of 2-Year Disease Recurrence in Anal Squamous Cell Carcinoma

**DOI:** 10.3390/cancers13020193

**Published:** 2021-01-07

**Authors:** Nicolas Giraud, Olivier Saut, Thomas Aparicio, Philippe Ronchin, Louis-Arnaud Bazire, Emilie Barbier, Claire Lemanski, Xavier Mirabel, Pierre-Luc Etienne, Astrid Lièvre, Wulfran Cacheux, Ariane Darut-Jouve, Christelle De la Fouchardière, Arnaud Hocquelet, Hervé Trillaud, Thomas Charleux, Gilles Breysacher, Delphine Argo-Leignel, Alexandre Tessier, Nicolas Magné, Meher Ben Abdelghani, Côme Lepage, Véronique Vendrely

**Affiliations:** 1Radiation Oncology Department, Hôpital Haut-Lévêque, CHU Bordeaux, 33600 Pessac, France; thomas.charleux@chu-bordeaux.fr (T.C.); veronique.vendrely@chu-bordeaux.fr (V.V.); 2Modelisation in Oncology (MOnc) Team, INRIA Bordeaux-Sud-Ouest, CNRS UMR 5251 and Université de Bordeaux, 33400 Talence, France; olivier.saut@inria.fr; 3CHU St Louis, 75010 Paris, France; thomas.aparicio@aphp.fr; 4Centre Azuréen de Cancérologie, 06250 Mougins, France; ronchinp@yahoo.fr; 5Institut Curie, 75005 Paris, France; louis.bazire@curie.fr; 6Fédération Francophone de Cancérologie Digestive, 21000 Dijon, France; emilie.barbier@u-bourgogne.fr; 7Institut du Cancer de Montpellier, 34090 Montpellier, France; claire.lemanski@icm.unicancer.fr; 8Centre Oscar Lambret, 59000 Lille, France; x-mirabel@o-lambret.fr; 9Centre Armoricain de Radiothérapie, 22190 Plérin, France; pl.etienne@wanadoo.fr; 10CHU Rennes, 35000 Rennes, France; astrid.lievre@chu-rennes.fr; 11Hôpital Privé Pays de Savoie, 74100 Annemasse, France; w.cacheux@ramsaygds.fr; 12Institut de Cancérologie de Bourgogne, 21000 Dijon, France; ariane.jouve@orange.fr; 13Centre Léon Bérard, 69008 Lyon, France; christelle.delafouchardiere@lyon.unicancer.fr; 14Radiology Department, CHU Bordeaux, 33000 Bordeaux, France; arnaud.hocquelet@chu-bordeaux.fr (A.H.); herve.trillaud@chu-bordeaux.fr (H.T.); 15Hôpitaux Civils de Colmar, 68000 Colmar, France; gilles.breysacher@ch-colmar.fr; 16GH Bretagne Sud, 56100 Lorient, France; d.argo-leignel@ghbs.bzh; 17CH Annecy, 74370 Annecy, France; atessier@ch-annecygenevois.fr; 18Institut de Cancérologie Lucien Neuwirth, 42270 St-Priest-en-Jarez, France; nicolas.magne@icloire.fr; 19Institut de Cancérologie Strasbourg Europe, 67200 Strasbourg, France; m.ben-abdelghani@icans.eu; 20CHU Dijon, 21000 Dijon, France; come.lepage@u-bourgogne.fr

**Keywords:** radiomics, machine learning, anal cancer, prediction medicine, precision medicine, magnetic resonance imaging

## Abstract

**Simple Summary:**

Exclusive chemo-radiotherapy (CRT) is the standard treatment for non-metastatic anal squamous cell carcinomas. Identifying novel prognostic factors could help to improve CRT outcomes, notably for locally advanced diseases where relapses still occur in around 35% of patients. In this study, we aim to assess the potential value of a pre-therapeutic MRI radiomic analysis added to standard clinical variables in order to build a logistic regression model predicting 2-year recurrence after CRT. In a population of 82 patients randomly divided in training (*n* = 54) and testing (*n* = 28) sets, after selection of optimal variables, a model using two radiomic (FirstOrder_Entropy and GLCM_JointEnergy) and two clinical (tumor size and CRT length) features was able to predict the 2-year recurrence with good performances in the testing set. Radiomic biomarkers provided valuable additional and independent information added to clinical data, and could help contribute to identify high risk patients amenable to treatment intensification with view of personalized medicine.

**Abstract:**

Purpose: Chemo-radiotherapy (CRT) is the standard treatment for non-metastatic anal squamous cell carcinomas (ASCC). Despite excellent results for T1-2 stages, relapses still occur in around 35% of locally advanced tumors. Recent strategies focus on treatment intensification, but could benefit from a better patient selection. Our goal was to assess the prognostic value of pre-therapeutic MRI radiomics on 2-year disease control (DC). Methods: We retrospectively selected patients with non-metastatic ASCC treated at the CHU Bordeaux and in the French FFCD0904 multicentric trial. Radiomic features were extracted from T2-weighted pre-therapeutic MRI delineated sequences. After random division between training and testing sets on a 2:1 ratio, univariate and multivariate analysis were performed on the training cohort to select optimal features. The correlation with 2-year DC was assessed using logistic regression models, with AUC and accuracy as performance gauges, and the prediction of disease-free survival using Cox regression and Kaplan-Meier analysis. Results: A total of 82 patients were randomized in the training (*n* = 54) and testing sets (*n* = 28). At 2 years, 24 patients (29%) presented relapse. In the training set, two clinical (tumor size and CRT length) and two radiomic features (FirstOrder_Entropy and GLCM_JointEnergy) were associated with disease control in univariate analysis and included in the model. The clinical model was outperformed by the mixed (clinical and radiomic) model in both the training (AUC 0.758 versus 0.825, accuracy of 75.9% versus 87%) and testing (AUC 0.714 versus 0.898, accuracy of 78.6% versus 85.7%) sets, which led to distinctive high and low risk of disease relapse groups (HR 8.60, *p* = 0.005). Conclusion: A mixed model with two clinical and two radiomic features was predictive of 2-year disease control after CRT and could contribute to identify high risk patients amenable to treatment intensification with view of personalized medicine.

## 1. Introduction

Anal squamous cell carcinoma (ASCC) is a relatively rare yet increasing disease, reaching around 48,000 new cases worldwide in 2018 [1]. With a low rate of distant metastases, ASCC is usually amenable to loco-regional treatment. Definitive chemo-radiotherapy (CRT) is the current standard of care for non-metastatic disease based on the results of several phase III trials performed in the 1990s [2,3,4]. Since then, the improvement in radiation techniques as well as a better chemotherapy (CT) management has allowed incremental and continuous progress in disease control rates as well as reducing early and late side effects [5,6]. Nevertheless, whilst excellent for small T1-2 tumors, CRT results remain more modest for locally advanced ASCC with up to 35% local or distant recurrences, more than 80% of them occurring during the first two years [7,8,9]. Several trials with intensification strategies in terms of radiation dose or chemotherapy association have been instigated but struggled to showcase clinically significant improvements compared to the standard association, perhaps due to non-optimal patient selections [10]. Recent efforts aim to escalate the treatment by adding targeted therapies, epidermal growth factor receptor (EGFR) inhibitors, to classic CRT, showing conflicting results. Among these trials, the French phase 1–2 FFCD-0904 trial added panitumumab, an anti-EGFR drug, to standard CRT, but the expected improvement in terms of complete pathological control at 8 weeks was not met [11,12,13,14]. Novel ongoing trials involving immunotherapy seem promising, but phase III results are yet to be presented [15]. Ideally, high-risk patients could be identified at baseline in order to offer more aggressive strategies, with regards to personalized medicine.

Radiomics is a newly emerging field of study, using data-characterization algorithms and mathematical tools applied on manually or automatically segmented volumes of interest on medical images [16,17]. After several steps, each bearing its own specifications (image acquisition, data standardization, segmentation, features extraction and qualification), a great number of quantitative metrics are extracted (Figure 1). These metrics individually describe various properties of the image, ranging from first-order attributes describing voxels’ intensities or their spatial distribution to higher-order features depicting for example the relationship between two or more adjacent voxels. The subsequent step is to integrate these new quantitative items inside predictive algorithms, for instance logistic regression models, alongside more conventional data (clinical, pathological, radiological, etc). The addition of these radiomic indicators display promising results in the field of oncology for a wide range of diseases [18,19]. In ASCC specifically, few papers have been published, using mostly PET or MRI imaging, and often small cohort sizes [20].

In this study, we aimed to assess the post-CRT 2-year disease-free survival (DFS) predictive performance of a logistic regression model incorporating MRI-based pre-CRT radiomics and clinical features.

## 2. Results

### 2.1. Patients Characteristics

Between January 2010 and September 2019, 86 patients underwent exclusive CRT for non-metastatic ASCC at the Bordeaux University Hospital. Among them, six patients were treated with suboptimal radiation dose because of palliative intent and 2 received an induction CT, thus excluded. In the end, 36 patients with an available pre-therapeutic MRI and a follow-up longer than 2 years were included.

Simultaneously, patients from the FFCD-0904 trial were included in the cohort. Of the 55 patients included in the phases I and II of this trial, four did not receive an optimal dose of CRT (toxicity assessment in phase I) and were excluded. Among the remaining patients, 46 had an available pre-CRT MRI.

In the end, a total of 82 patients were selected for further examination. Patient selection is detailed in Figure 2. Clinical and histopathological did not significantly differ between the two cohorts, subsequently blended (Table 1). Patients from the joint cohort were randomly divided into training (*n* = 54) and testing (*n* = 28) sets.

### 2.2. Outcome

Median follow-up in the event-free population was 24 months (range, 24–101). Among the 82 patients, loco-regional or distant relapse occurred in 24 patients (29%), after a median duration of 5 months (1–14 months). Among them, 12 patients (50%) presented with in-field relapse, eight (33%) with metastatic-only relapse and four (17%) with both. Nine patients died after tumor relapse (median 11 months, range 1–40 months).

### 2.3. Logistic Regression

Using univariate analysis, seven variables (two clinical and five radiomic) were associated with the outcome in the training set: tumor length (*p* = 0.01), CRT duration (*p* = 0.043), FO_Entropy (*p* = 0.045), FO_Uniformity (*p* = 0.045), Glcm_JointEnergy (*p* = 0.047), Glcm_JointEntropy (*p* = 0.049) and Glrlm_GrayLevelVariance (*p* = 0.045). Description of these radiomic features can be found in Table 2.

Strongly intercorrelated features were investigated using a correlation matrix and Pearson tests. FO_Uniformity was found to be strongly correlated with Glrlm_GrayLevelVariance and FO_Entropy; and Glcm_JointEnergy with Glcm_JointEntropy (Figure 3). Among these, FO_Entropy for the first group and Glcm_JointEnergy for the second were most correlated with the outcome, thus retained in the model alongside the tumor length and CRT duration. In multivariate analysis, adding to these four variables the patients’ age and sex, the tumor length (*p* = 0.036) and Glcm_JointEntropy (*p* = 0.026) remained significantly associated with the 2-year DFS.

ROC curves were drawn in order to determine the optimal classification threshold for each of the four model’s variables, using the Youden method (Table 3). These thresholds allowed to dichotomize each of the four quantitative features and were applied to both the training and testing sets. A binary logistic regression model was subsequently fitted on the training set based on these four dichotomized features, and performed 2-year DFS predictions in the two sets.

In the training set, this mixed radiomic and clinical model obtained an accuracy of 87%, with an area under the curve (AUC) of the ROC curve of 0.825 (sensitivity of 58.8%, specificity of 100% to predict the recurrence). In the testing set, a prediction accuracy of 85.7% was achieved, with a ROC-AUC of 0.898 (sensitivity of 42.8%, specificity of 100% to predict the recurrence).

It allowed to distinguish two groups among the testing set according to the risk of recurrence after CRT, with a hazard ratio of 8.6 (*p* = 0.005), Figure 4. This model performed better in the testing set than the clinical-only (accuracy of 78.6%, ROC-AUC 0.714) and radiomics-only (accuracy of 67.9%, ROC-AUC of 0.677) models.

### 2.4. Inter-Reader Variability

In the relevant sample (*n* = 22), the two independent segmentations were relatively similar, with a mean DICE coefficient of 0.87 and an average mean Hausdorff distance of 0.68mm. In this sample, the differences in segmentation did not influence the model’s predictions.

## 3. Discussion

Male gender, N-positive stage and tumor length greater than 5 cm are among the few recognized ASCC clinical prognostic factors, associated with worse clinical outcomes [21,22]. Other prognostic factors have been suggested throughout the years, like an age greater than 55, increased circumferential tumor spread, skin ulceration, inguinal node development and total RT dose superior to 60 Gy for worse colostomy-free survival [23], or the baseline neutrophil to lymphocyte ratio for loco-regional recurrence [24]. Prognostic nomograms have even been elaborated, with promising results in predicting cancer specific survival and overall survival as well as risk-stratifying patients, but were only based on clinical data [25]. In that regard, radiomics features extracted from medical imaging could bring additional informative data. Adding these novel parameters to previously used clinical parameters seems to offer additive performance, paving the way to build more accurate predictive models, but have for now barely been explored in ASCC [20].

In this multicentric study, the largest to our knowledge to explore the input of MRI T2w radiomics in ASCC recurrence prediction, we showed that a simple mixed (clinical and radiomic) LR model was able to predict the 2-y DFS with good performances using optimal thresholds determined on the training set. This model incorporates two clinical and two radiomics features. Among the two studies published on this subject, Hocquelet et al. found that Skewness and Cluster Shade were correlated with recurrence or death, which we did not highlight in our study [26]. Owczarczyk et al. observed that baseline T2w Energy, a witness of voxel signal distribution (high energy being linked to homogeneous intensity distribution), was associated with 2-year DFS [27]. In our study, although FO_Energy was not statistically associated with the outcome (*p* = 0.38 in the training set), FO_Entropy and FO_Uniformity also depict to some extent the heterogeneity of voxels’ intensities [28]. In both trials, the homogeneity in baseline tumor voxel intensities seems to be linked to poorer outcomes.

In our LR model, thresholds were used in order to dichotomize the quantitative variables of interest using optimal values given by the Youden method on ROC curves. While this raises a possible “threshold effect”, the value found for tumor length was 49 mm, which is consistent with the literature data (50 mm usually considered as a critical prognostic threshold and the boundary between T2 and T3 in the TNM 8th edition). This value is more difficult to interpret for radiomic variables as their absolute values are harder to apprehend. Moreover, while showcasing overall good performances on prediction accuracy, specificity and ROC AUC, the results for sensitivity are poorer (42.9%), implying a possible high number of false negatives. Consequently, while reducing the risk of over-treatment (all patients predicted to present disease recurrence relapsed), the cost would be a few patients that would exhibit disease recurrence not being pre-detected (under-diagnosis). This raises a fundamental question, as one could argue that with the current CRT results in locally advanced ASCC, the priority should be given to detecting all recurring patients, even if some other patients could be over-treated. Thirdly, this model takes into account the length of CRT (removing it resulted in poorer performances), which de facto prevents from using this model in intensification trials including adjuvant chemotherapy or dose escalation. However, it highlights an important message that avoiding as much treatment interruptions as possible and reducing their duration seem to improve the patients’ outcome, pauses being increasingly questioned [29]. As a matter of fact, in a recent pooled analysis, shorter overall treatment time was associated with longer PFS [30]. Finally, a significant proportion of patients who entered in this model received an experimental drug (panitumumab) not standardly used, but this was taken into account in the model and did not seem to interfere with the outcome.

In this study, we limited our specter of imaging to T2w sequences because other modalities were unsystematically performed, but inclusion of other imaging modalities e.g., diffusion MRI sequences or PET-CT imaging could further improve the model’s predictive performances. Moreover, the benefit of adding pathological data into the model, such as the p16 status obtained on the initial biopsy or the measurement of tumor infiltrating lymphocytes (TILs), of probable prognostic value, could warrant further investigations [31,32,33].

## 4. Materials and Methods

### 4.1. Patient Selection

All patients treated by CRT at the CHU Bordeaux between 2010 and 2019 for histologically proven non-metastatic ASCC were retrospectively considered. All patients treated with palliative intent or who received induction CT were excluded, as well as those event-free with less than 2 years of follow-up.

Additionally, patients from the French FFCD-0904 phase I-II multicentric prospective study (NCT01581840) were included in the cohort. This trial included patients with T2 > 3 cm, T3-4 and/or N+ non-metastatic tumors. All participating patients signed an informed consent. The study protocol was approved by the Ethics Committee at site. The study was conducted according to the principles of the Declaration of Helsinki, the International Conference on Harmonization Guideline on Good Clinical Practice, the French laws and regulations.

All patients for which the pre-therapeutic MRI could not be retrieved were excluded.

### 4.2. Outcome

The primary endpoint was the 2-year DFS, relapse being defined as clinical or radiological loco-regional or distant progression, either directly after the CRT course (CRT resistance) or after an initial response. In doubtful cases, histology was mandatory.

### 4.3. MRI

The images were performed on different 1.5 or 3-T MRI machines, with protocols left to the choice of each expert center in order to adhere as much as possible to real-life conditions. Patients were scanned in supine position. Analyzed MRI sequences included axial T2-weighted (T2w) sequences, without fat suppression. We chose to focus on the texture analysis of T2w images because they are easy to acquire, less prone to artefacts, non-invasive and suitable for every patient. Importantly, T2w sequences offer a high signal-to-noise ratio, spatial resolution and soft tissue contrast images of the anal sphincter structures.

### 4.4. Clinical Features and Treatment-Related Data

The following clinical variables were collected from medical records: age, sex, HIV status, TNM stage, pre-CRT tumor length, pre-rectal septum infiltration. When applicable, date and type (local, distant or both) of relapse was recorded. All categorical clinical features were remapped to ordinal values and binarized.

### 4.5. Chemo-Radiotherapy

All patients underwent CRT as modality of treatment using at least 6 MV photons. The total duration of the CRT (total number of days between the first and final days of RT, treated as a continuous variable), the received radiation dose and the type of CT (including the prescription of panitumumab) were congregated.

In the Bordeaux cohort, radiation doses were 59.4 Gy in 33 fractions for all patients. CT consisted of Xeloda (1650 mg/m^2^ on RT treatment days) and mitomycin C (MMC, two injections of 10 mg/m^2^), 5-Fluorouracil (5FU, 800 mg/m^2^/day over 4 days) and Cisplatin (80 mg/m^2^) or Xeloda alone (1650 mg/m^2^ on RT treatment days).

In the FFCD cohort, all patients received a 65 Gy radiation dose in two consecutive sequences (45 Gy in 25 fractions followed by 20 Gy in 10 fractions), with no treatment pause. The CT protocol associated four injections of panitumumab 3 mg/kg and two injections of continuous 5FU 400 mg/m^2^/day over 4 days plus MMC 10 mg/m^2^, as defined by the phase I toxicity assessment [14].

### 4.6. Tumor Delineation

Anal canal tumors were manually delineated on all axial slices using Varian Eclipse^®^ on pre-therapeutic T2w sequences and validated by two experts. A second segmentation was blindly performed by one expert for 27% (*n* = 22) patients, allowing to compute concordance indices based on DICE coefficients and average Hausdorff distances [34].

### 4.7. MRI Presampling

Prior to radiomic extraction, transformations were applied to the pre-CRT T2w sequences. A N4 bias correction algorithm was carried out based on the simple itk python library to correct MRI low frequency intensity non-uniformity. Then, an in-house python script was created to normalize intra-VOI voxel intensities.

### 4.8. Radiomic Features

A total of 100 radiomics quantitative features were extracted from each segmented volume using the PyRadiomics library, including first-order (FO) shape and intensity indicators as well as second-order textural features based on the gray level co-occurrence (GLCM), gray level run length (GLRLM), gray level size zone (GLSZM) and gray level dependence (GLDM) matrixes.

### 4.9. Statistical Analysis and Feature Selection

The cohort was first randomly split into two sets, 2/3 for training (*n* = 54) and 1/3 for testing (*n* = 28). Clinical and radiomic variables (listed in Appendix A) were assessed for their 2-year DFS predictive performance with univariate (Student if applicable or Wilcoxon tests for continuous variables and Fisher tests for categorical features, as well as receiver operating characteristic (ROC) curves) and multivariate (logistic regression) analyses.

These analyses in the training set allowed feature selection. After eliminating redundant variables using the Pearson correlation test (when two variables were significantly correlated, the variable most correlated with the outcome was chosen), optimal cut-off values for each elected variable were defined using the Youden method applied on each ROC curve. This threshold was then applied in the two sets. Quantitative performance evaluation was carried out using accuracy, sensitivity, and specificity regarding prediction of relapse and the log-rank test regarding DFS stratification. All statistical analyses were carried out using the RStudio software (v1.3.959).

## 5. Conclusions

After optimization and performance assessment, a four-variable model fitted with a LR algorithm achieved good performances to predict 2-y DFS. With a 100% positive predictive value, this model could help identifying patients at very high risk of recurrence, allowing for a better guidance of patients eligible to treatment intensification (adjuvant CT, or RT dose escalation for example). The ability to foretell relapse could guide therapeutic strategy and lead to intensification on a personalized basis. Adding radiomic features to clinical data improved the model’s results, individualizing some patients with a very high risk of relapse. Yet, another portion of patients who eventually relapsed were not identified by the model, illustrating their poor discrimination compared to relapse-free patients according to the existing clinical and radiomic features.

As Machine Learning models thrive with high-dimensional data, new biomarkers as well as other imaging modalities or repeated imaging obtained throughout the treatment for delta-radiomics (for instance with novel MRI linear accelerators) could further bring relevant information to the model. In addition, exploring the correlation between these features and the outcome with a longer follow-up is warranted, as well as including more patients. This is why there is a strong need of national or international collaboration, especially in neoplasms like ASCC with relatively low incidences. The next intended step is to challenge such models in a prospective setting, before considering using them in interventional trials.

## Figures and Tables

**Figure 1 cancers-13-00193-f001:**
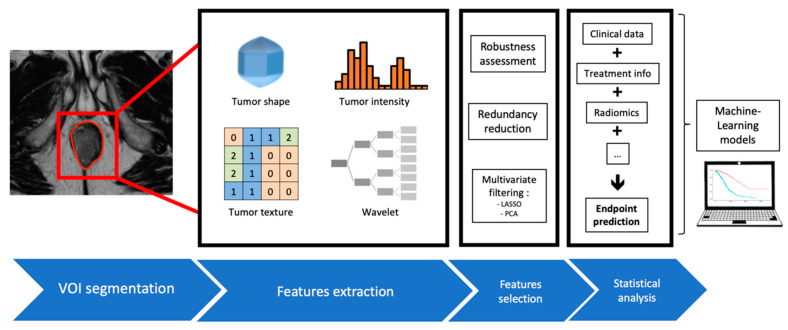
Radiomics workflow and integration into Machine Learning models.

**Figure 2 cancers-13-00193-f002:**
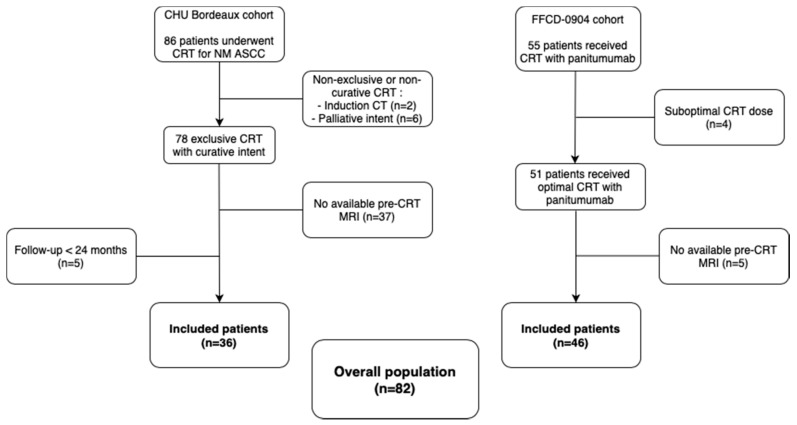
Flowchart of patient selection. (Abbreviations: CRT = chemo-radiotherapy, NM = non-metastatic, ASCC = anal squamous cell cancer, CT = chemotherapy, MRI = magnetic resonance imaging).

**Figure 3 cancers-13-00193-f003:**
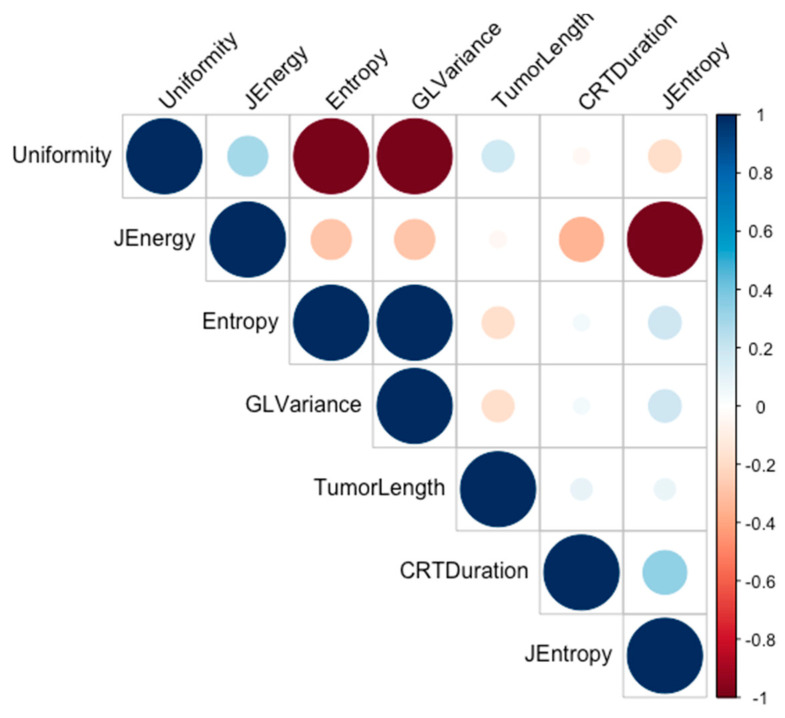
Correlation matrix between features significantly associated with the outcome in the training cohort. (Abbreviations: JEnergy = Glcm_JointEnergy, JEntropy = Glcm_JointEntropy, GLVariance = Glrlm_GrayLevelVariance, CRTDuration = Chemo-radiotherapy duration).

**Figure 4 cancers-13-00193-f004:**
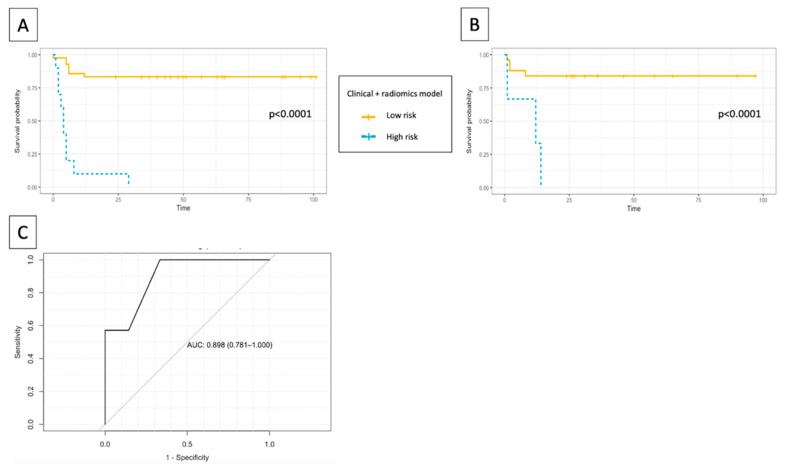
Kaplan–Meier estimates of disease-free survival using the combined (clinical and radiomics) model for (**A**) training and (**B**) testing set, and ROC curve in the testing set (**C**).

**Table 1 cancers-13-00193-t001:** Patient and tumor characteristics in the two subsequently blended cohorts. (Abbreviations: F = female, M = male, HIV = human immunodeficiency virus, DFS = disease-free survival).

	Bordeaux Cohort (*n* = 36)	FFCD Cohort (*n* = 46)	*p*-Value
Age at diagnosis (mean, years)	62.3	60.1	0.39
Sex:			0.96
- F	30/36 (83.3%)	37/46 (80.4%)	
- M	6/36 (16.7%)	9/46 (19.6%)	
HIV status:			0.79
- Positive	1/36 (2.8%)	3/46 (6.5%)	
- Negative	35/36 (97.2%)	43/46 (93.5%)	
Tumor length (mean, mm)	51.2	47.5	0.40
Pre-rectal tumor infiltration	16/36 (44.4%)	13/46 (28.2%)	0.20
T-stage:			0.44
- cT1-2	11/36 (30.6%)	19/46 (41.2%)	
- cT3-4	25/36 (69.4%)	27/46 (58.7%)	
N-stage:			1
- cN0	10/36 (27.8%)	12/46 (26.1%)	
- cN+	26/36 (72.2%)	34/46 (73.9%)	
AJCC groupings:			0.20
- I	1/36 (2.8%)	0/46 (0%)	
- IIa	4/36 (11.1%)	5/46 (10.9%)	
- IIb	1/36 (2.8%)	4/46 (8.7%)	
- IIIa	6/36 (16.7%)	14/46 (30.4%)	
- IIIb	4/36 (11.1%)	3/46 (6.5%)	
- IIIc	20/36 (55.5%)	20/46 (43.5%)	
2-year DFS	23/36 (63.9%)	35/46 (76.1%)	0.34

**Table 2 cancers-13-00193-t002:** Signification of the radiomics features significantly associated with the outcome in the training cohort.

FO_Uniformity	$\overset{Ng}{\underset{i=1}{\sum }}p{\left(i\right)}^{2}$	Sum of the squares of each intensity value. A greater uniformity implies a greater homogeneity or a smaller range of discrete intensity values.
FO_Entropy	$-\overset{Ng}{\underset{i=1}{\sum }}p\left(i\right)log2\left(p\left(i\right)+\varepsilon \right)$	Randomness in the image values, with *ϵ* an arbitrarily small positive number (≈2.2 × 10^−16^).
GLCM_JointEnergy	$\overset{Ng}{\underset{i=1}{\sum }}\overset{Ng}{\underset{j=1}{\sum }}(p{\left(i,j\right)}^{2})$	Measure of homogeneous patterns in the image. A greater energy implies more intensity value pairs that neighbor each other at higher frequencies.
GLCM_JointEntropy	$-\overset{Ng}{\underset{i=1}{\sum }}\overset{Ng}{\underset{j=1}{\sum }}p\left(i,j\right)log2\left(p\left(i,j\right)+\varepsilon \right)$	Measure of the variability in neighborhood intensity values.
GLRLM_GrayLevelVariance	$\overset{Ng}{\underset{i=1}{\sum }}\overset{Ns}{\underset{j=1}{\sum }}p\left(i,j\right){\left(i-\mu \right)}^{2}$	Variance in gray level intensities for the zones.

**Table 3 cancers-13-00193-t003:** Correlation between variables and 2-year disease recurrence. (Abbreviations: Se = sensitivity, Sp = specificity).

Variable	Univariate Analysis			Best Cut-Off	*p*-Value	Multivariate Analysis	
	AUC	Se (%)	Sp (%)			Odds-Ratio	*p*-Value
Tumor length (mm)	0.71 (0.56–0.86)	71	65	49	0.01	5.75	0.036
CRT duration (days)	0.67 (0.52–0.83)	82	51	50.5	0.043		
FO_Entropy	0.67 (0.51–0.84)	59	87	0.992	0.045	0.12	0.026
GLCM_JointEnergy	0.67 (0.50–0.84)	41	95	0.317	0.047		

## Data Availability

The data presented in this study are available on request from the corresponding author. The data are not publicly available due to privacy reasons (individual patient data, partly belonging to a FFCD academic trial unpublished to date).

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
