# Peer review of "MRI-Based Radiomics Input for Prediction of 2-Year Disease Recurrence in Anal Squamous Cell Carcinoma"

_cancers, 2021, doi:10.3390/cancers13020193_

Round 1
Reviewer 1 Report
This is an interesting and novel investigation of radiomic parameters as prognostic variables with respect to chemoradiotherapy for ASCC. In general it is well performed and described and although small numbers make this hypothesis-generating as opposed to practice-defining, it represents interesting data that readers will be stimulated to investigate.
I have a number of suggestions that would improve the manuscript:
Introduction second sentence (p2,line 70) - ASCC doesn't have an indolent natural history, just that spread is more usually locoregional than metastatic.
Results table 1 - please give AJCC groupings also to help understand the patient cohort
Results - please give the details around CRT duration in the population - and was this handled as a continuous variable (even though the majority must be at a standard number of days) or as days > expected?
Results - logistic regression - give details on which variables were tested to generate the 7 'significant' ones that made it to the model and give the results of all variables somewhere in the data
Discussion - there is more to say around biological features that are associated with outcomes following chemoRT - in particular around HPV status where p16 negative patients do worse and also where immune recognition e.g. TILs also have prognostic ability (Gilbert et al Radiotherapy and Oncology 2013, Serup-Hansen et al JCO 2013, reviewed in Jones et al BJC 2017).
Discussion - how to build a model including biological and radiomic features?
Discussion - more detail please around how length of CRT impacts on the model. It is implied that removing this factor limits the usefulness of the model? Please explain this - how many patients had extended treatment times? Chance or biological reasons for this?
Author Response
Dear reviewer,
We thank you very much for your careful reading and feedback regarding our manuscript.
To answer the different points raised:
- Results table 1 - please give AJCC groupings also to help understand the patient cohort
We added the AJCC groupings to Table 1 as suggested.
- Results - please give the details around CRT duration in the population - and was this handled as a continuous variable (even though the majority must be at a standard number of days) or as days > expected?
The CRT duration was defined as the total number of days between the first and last days of RT, treated as a continuous variable. We detailed a bit more the radiation protocol (notably the number of fractions delivered) in the corrected version of the manuscript.
- Results - logistic regression - give details on which variables were tested to generate the 7 'significant' ones that made it to the model and give the results of all variables somewhere in the data
We added the complete list of clinical and radiomic variables tested in a Supplementary Table.
- Discussion - there is more to say around biological features that are associated with outcomes following chemoRT - in particular around HPV status where p16 negative patients do worse and also where immune recognition e.g. TILs also have prognostic ability (Gilbert et al Radiotherapy and Oncology 2013, Serup-Hansen et al JCO 2013, reviewed in Jones et al BJC 2017).
p16 status was not available for all patients because we didn’t have access to all initial pathological reports, especially for FFCD0904 patients. It could nevertheless be very interesting to add them to the model, alongside TILs for example as suggested, as they could bring additional value to a model already combining clinical and radiomic data. A sentence was added in our discussion to reflect that perspective.
- Discussion - more detail please around how length of CRT impacts on the model. It is implied that removing this factor limits the usefulness of the model? Please explain this - how many patients had extended treatment times? Chance or biological reasons for this?
Indeed, we found that removing the CRT length variable to our logistic regression model decreased its predictive performances thus we chose to keep it in our final model. In the phase 2 FFCD-0904 trial (to be published soon), around 30% patients had RT interruptions with a median duration of 6 days (2-15 days), mostly due to toxicity. Moreover, in some centers, treatment pauses after the first 45Gy pelvic irradiation are still performed, thus why we chose to insist on the importance of avoiding as much as possible unessential interruptions.
Reviewer 2 Report
The notion of individualised radiotherapy treatment in anal SCC is clinically relevant and currently being explored in clinical trials. At present, patient selection for dose escalation/ de-escalation is performed based on clinical indices, most importantly tumour size and node involvement. In the last few years, there have been attempts to use quantitative imaging biomarkers to improve patient stratification.
Several retrospective studies and more recently, two prospective trials (Jones et al; Muirhead et al) have assessed the diagnostic performance of mixed clinico-imaging models and found them superior to clinical models alone.
Giraud et al have carried out a retrospective radiomic analysis of pre-treatment T2w MRIs and identified two imaging parameters namely First Order Entropy and GLCM_JointEnergy associated with increased risk of disease recurrence/ 2y-DFS. The association remained significant when corrected for clinical variables (tumour length and CRT duration).
This is an important and topical study. I am sure it would be of interest to "Cancers" readers.
I have some minor comments regarding methodology and conclusions:
1.In terms of patient selection for the training and testing set, I would appreciate a clarification of the rationale behind blending the two investigated cohorts rather than using one cohort as the training set (e.g. FFCD cohort ) and the other as test set (Bordeoux cohort) which would have provided a more robust assessment of the biomarker performance (as they constitute two independent cohorts and could be perceived as a form of external validation)
2. With median time to recurrence of 5 months - are the events occurring in patients with primary disease progression rather than disease recurrence - it would be useful to make this distinction in, both, the outcomes methods section as well as elaborated on in the discussion.
It would help the readers understand whether the proposed model has the ability to identify primary RT resistant tumours (for instance large and necrotic) or whether it can also identify tumours that initially respond but have an increased risk of subsequent recurrence. This would potentially have treatment implications (CRT intensification/ use of radiosensitisers or RT+IO versus more assiduous follow-up with salvage surgery when required).
3. In terms of discussion, it would be useful quantify how the radiomic biomarkers improve the performance of the already established clinical (T stage = 5cm cut-off) biomarker - I would like to see the odds of recurrence in patients with cT2 versus cT3 disease (tumour length > 5cm) when stratified by radiomic biomarker thresholds - for me as a clinician that would represent the real value of the proposed model ie identifying high risk cT2 tumours and low risk cT3 tumours with a real chance to alter management and improve outcomes and QoL.
Minor points:
Line 76 despite should be: whilst
Author Response
Dear reviewer,
Thank you for your remarks and propositions.
To answer your observations:
1- In terms of patient selection for the training and testing set, I would appreciate a clarification of the rationale behind blending the two investigated cohorts rather than using one cohort as the training set (e.g. FFCD cohort ) and the other as test set (Bordeoux cohort) which would have provided a more robust assessment of the biomarker performance (as they constitute two independent cohorts and could be perceived as a form of external validation)
The reasoning behind the cohort blending lays upon a few points:
- We wanted to include more patients in the training cohort, as we felt more information could be derived for the model from a larger number of patients in the training set.
- We wanted to avoid a difference of outcomes based solely on the treatment received (e.g. adding panitumumab in the FFCD cohort), which was partly taken into account by randomizing patients between the training and testing sets.
- A significant proportion of patients belonging to the FFCD cohort were treated in the CHU Bordeaux.
We however hope to conduct an external validation in a prospective setting.
2- With median time to recurrence of 5 months - are the events occurring in patients with primary disease progression rather than disease recurrence - it would be useful to make this distinction in, both, the outcomes methods section as well as elaborated on in the discussion.
The definition of PFS was based on the progression on the first MRI post-CRT or after an initial response (partial or complete). Only 2 patients had already apparent progressive disease at 6 weeks post-CRT, the rest had at least a partial response on the first MRI evaluation. Nevertheless, as a clear tumor response assessment is not easy in the first controls and can be delayed, as well as given the low number of patients it was difficult to clearly differentiate primary RT resistant and true recurrences in our cohort.
3- In terms of discussion, it would be useful quantify how the radiomic biomarkers improve the performance of the already established clinical (T stage = 5cm cut-off) biomarker - I would like to see the odds of recurrence in patients with cT2 versus cT3 disease (tumour length > 5cm) when stratified by radiomic biomarker thresholds - for me as a clinician that would represent the real value of the proposed model ie identifying high risk cT2 tumours and low risk cT3 tumours with a real chance to alter management and improve outcomes and QoL.
This is a very good point and could definitely be explored as it could indeed bring additional help to the therapeutic decision. In our cohort, given the relatively low number of patients and events, it is difficult to stratify by stage and extrapolate with sufficient statistical power.